# The differential impact of pediatric COVID-19 between high-income countries and low- and middle-income countries: A systematic review of fatality and ICU admission in children worldwide

**Taito Kitano**[1]*, **Mao Kitano**[2], **Carsten Krueger**[1], **Hassan Jamal**[1], **Hatem Al Rawahi**[1], **Rachelle Lee-Krueger**[3], **Rose Doulin Sun**[1], **Sandra Isabel**[1], **Marta Taida García-Ascaso**[1], **Hiromi Hibino**[4], **Bettina Camara**[5], **Marc Isabel**[6], **Leanna Cho**[1], **Helen E. Groves**[1], **Pierre-Philippe Piché-Renaud**[1], **Michael Kossov**[7], **Ikuho Kou**[8], **Ilsu Jon**[9], **Ana C. Blanchard**[1], **Nao Matsuda**[10], **Quenby Mahood**[11], **Anupma Wadhwa**[1], **Ari Bitnun**[1], **Shaun K. Morris**[1,12]

**1** Department of Pediatrics, The Hospital for Sick Children, University of Toronto, Toronto, ON, Canada, **2** Mikage Child Dental Clinic, Kobe, Hyogo, Japan, **3** Faculty of Education, University of Ottawa, Ottawa, ON, Canada, **4** McEwen Stem Cell Institute, Universal Health Network, Toronto, ON, Canada, **5** Faculty of Immunology, Department of Medicine, University of Toronto, Toronto, ON, Canada, **6** Département de mathématique, Faculté des sciences et génie, Université Laval, Pavillon Alexandre-Vachon, Québec, QC, Canada, **7** Department Laboratory Medicine and Pathobiology, Division of Medical Microbiology, The Hospital for Sick Children at University of Toronto, Toronto, ON, Canada, **8** Kaji Dental Clinic, Kobe, Hyogo, Japan, **9** Utsunomiya Kyoritsu Clinic, Utsunomiya, Tochigi, Japan, **10** Alpaca Child Dental Clinic, Hiroshima, Hiroshima, Japan, **11** Hospital Library and Archives, Learning Institute, The Hospital for Sick Children at University of Toronto, Toronto, ON, Canada, **12** Centre for Global Child Health and Child Health Evaluative Sciences, The Hospital for Sick Children, Toronto, ON, Canada

* taito.kitano@sickkids.ca

## Abstract

### Background

The overall global impact of COVID-19 in children and regional variability in pediatric outcomes are presently unknown.

### Methods

To evaluate the magnitude of global COVID-19 death and intensive care unit (ICU) admission in children aged 0–19 years, a systematic review was conducted for articles and national reports as of December 7, 2020. This systematic review is registered with PROSPERO (registration number: CRD42020179696).

### Results

We reviewed 16,027 articles as well as 225 national reports from 216 countries. Among the 3,788 global pediatric COVID-19 deaths, 3,394 (91.5%) deaths were reported from low- and middle-income countries (LMIC), while 83.5% of pediatric population from all included countries were from LMIC. The pediatric deaths/1,000,000 children and case fatality rate (CFR)

**Data Availability Statement:** All relevant data are within the manuscript and its Supporting Information files.

**Funding:** The authors received no specific funding for this work.

**Competing interests:** The authors have declared that no competing interests exist.

were significantly higher in LMIC than in high-income countries (HIC) (2.77 in LMIC vs 1.32 in HIC; *p* < 0.001 and 0.24% in LMIC vs 0.01% in HIC; p < 0.001, respectively). The ICU admission/1,000,000 children was 18.80 and 1.48 in HIC and LMIC, respectively (*p* < 0.001). The highest deaths/1,000,000 children and CFR were in infants < 1 year old (10.03 and 0.58% in the world, 5.39 and 0.07% in HIC and 10.98 and 1.30% in LMIC, respectively).

## Conclusions

The study highlights that there may be a larger impact of pediatric COVID-19 fatality in LMICs compared to HICs.

## Introduction

Since December 2019, the Severe Acute Respiratory Syndrome (SARS) coronavirus 2 (SARS--CoV-2) has spread throughout the world and coronavirus disease 2019 (COVID-19) has resulted in more than 85 million cases and approximately 1.8 million deaths worldwide as of Dec 31, 2020 [1]. Case counts and deaths by country have been reported in near real time by several sources [1, 2].

Previous large cohort studies from many countries and review articles of children with COVID-19 concluded that death was rare [3–7]. However, the overall global impact of COVID-19 in children is presently unknown, as is how the impact for children varies between countries. Investigating the differential impact of COVID-19 in children by country is important to direct limited global resources to more vulnerable regions. The number of deaths and intensive care unit (ICU) admissions per capita as well as case fatality rate (CFR) and ICU admission rate due to COVID-19 have been widely used as measures of COVID-19 severity and are more likely to reflect true impact than number of cases which is dependent on local testing patterns and the size of population. Notably, less access to ICU level care in more resource limited countries results in an inability to provide the highest level care to the most critically ill which may be related to higher case fatality [8].

To better understand the global epidemiology of COVID-19 in children and differences in outcome across countries, we conducted a systematic review of multiple databases in multiple languages as well as national reports from governments or public health authorities.

## Methods

This study was conducted in accordance with Preferred Reporting Items for Systematic Reviews and Meta-Analysis (PRISMA) guidelines [9]. This systematic review is registered with PROSPERO (registration number: CRD42020179696). Our primary objective was to provide a robust global data of pediatric COVID-19 deaths and ICU admissions. Our secondary objectives were to determine pediatric CFR and ICU admission rates and to compare results across HIC and LMIC.

The case definition included children (aged 0–19 years) with polymerase chain reaction (PCR)-confirmed SARS-CoV-2 infection. Serologically diagnosed cases without PCR confirmation were excluded from our analysis. The targeted focus of this study is to evaluate the global impact of PCR-confirmed SARS-CoV-2 infection in children. The epidemiology of the multi-system inflammatory syndrome in children (MIS-C), including deaths and ICU

admissions from this condition, was excluded. Case reports, case series, prospective studies, case control studies, and cross-sectional studies were all eligible data sources.

## Search strategy

We searched MEDLINE, Embase, the Cochrane Library, CINAHL and WHO COVID-19 databases to identify articles related to or including any COVID-19 cases without language restriction. To more fully capture global data, we also searched the following non-English databases: CNKI and Wanfang (Chinese), Kmbase (Korean), ICHUSI Web (Japanese), LILAC and SciELO (Spanish and Portuguese), LiSSa (French), Ulakbim (Turkish), Magiran (Farsi), Islamic world citation center (Arabic) and Russian Scientific Electronic Library (Russian). The article search was completed in three times. The first search using all databases was conducted on April 30 (search 1) and the second and the third search using only MEDLINE and Embase was conducted on August 10 (search 2) and December 7 (search 3). The details of the search strategy are provided in S1 File.

## Study selection and risk of bias assessment

Articles were extracted from each database using Endnote X8 (Clarivate Analytics US LLC, PA), and then the software package COVIDENCE (Veritas Health Innovation; www. covidence.org) was used to manage these. Extracted studies were screened by two reviewers independently. Full text reviews of English and non-English articles were conducted by two independent reviewers fluent in the language of the study or report. Studies were screened using the inclusion and exclusion criteria described below. We included studies with any pediatric COVID-19 case (aged 0–19 years) from which fatality or ICU admission data could be extracted. We excluded studies without age-specific information, without any pediatric cases, and without any extractable outcome data for the pediatric cases. Review articles, guidelines, expert opinions, articles whose main topic was infection control, mathematical modelling studies, molecular studies, animal studies, studies about other coronaviruses, and overlapping datasets were also excluded. If more than one study from the same population was found (overlapping data), the study with the most comprehensive result or the largest pediatric sample size was included. For feasibility reasons, in search 3 we included articles reporting nationwide subgroup outcomes (neonatal, age-specific or ICU respiratory support outcomes) but excluded other articles from countries with comprehensive national reports for fatality and ICU outcomes. For articles with overlapping data, where one article had more comprehensive data for fatality and the other article had more comprehensive ICU admission data, both were included and the more comprehensive data was extracted from the appropriate paper. Additional methodologic details of dealing with overlapping data are provided in S1 File (page 18–19). Discrepancies in inclusion or exclusion during screening were discussed between the two reviewers until consensus was achieved or resolved by a third reviewer. In cases of ambiguity of data in extracted articles, for example if we could not rule out a possibility of overlap with another article or if we could not find clear outcomes of included cases (ICU admission or maximal respiratory support), the authors of the studies were contacted.

Two independent reviewers evaluated the quality of included studies using Critical Appraisal Tools of Joanna Briggs Institute [10]. Discrepancies in the risk of bias assessment tools were discussed between the two reviewers until consensus was achieved or resolved by a third reviewer.

## National report search

To assess grey literature sources, we followed the Canadian Agency for Drugs and Technology checklist [11] and searched health surveys, case notification systems, national reports or survey

data and governmental official reports from national Centers for Disease Control (CDC), governments, ministries of health, national departments of health or national academic associations in all 218 countries defined by the World Bank. All references from their websites were investigated to find national reports for pediatric COVID-19 cases and age-specific COVID-19 cases using the same inclusion and exclusion criteria for articles. The national report data searches were performed on May 22–24 (search 1), August 27-September 1 (search 2), and December 7–11, 2020 (search 3).

## Data extraction and analysis

A Microsoft Excel datasheet was used to track all included articles, governmental and public health reports. Data extraction forms were independently completed by each reviewer. For included studies and reports, we extracted the number, age and clinical outcomes (death and ICU admission) of all identified pediatric cases. For children admitted to ICU, if available, data on the type of ICU admission, maximal respiratory support and required vasopressor use were also extracted. Additional clinical data including laboratory results and directed treatments were not extracted. A third reviewer checked the article/report list and reviewed the included extraction forms of the first two reviewers to ensure accuracy, to ensure no duplication of articles or datasets, and to resolve any discrepancies. First, all pediatric COVID-19 deaths and ICU admissions were pooled to calculate the global and national number of deaths and ICU admissions in children. If a child was admitted to ICU and died due to COVID-19, we counted the case as one death and one ICU admission. Then, COVID-19 deaths or ICU admissions/1,000,000 children were calculated by dividing the number of pediatric COVID-19 deaths in each country by the total number of children (0-19years of age) in that population. National pediatric populations by age for all countries were obtained from data in United Nations (estimated population on July 1, 2020) [12]. Data including the number of pediatric SARS-CoV-2 infections and outcomes (pediatric death or ICU admission) were also synthesized to calculate the combined pediatric CFR and ICU admission rate. The data with both denominator (the number of confirmed SARS-CoV-2 infections) and numerator (the number of deaths or ICU admissions) were included in our calculation of CFR and ICU admission rate. The outcome data were pooled at a country level. We combined the number of pediatric cases and the number of pediatric outcome events from all data extracted to calculate the pooled pediatric CFR and ICU admission rate. The number of deaths and ICU admissions as well as CFRs and ICU admission rates were compared between high-income countries (HICs) and low- and middle-income countries (LMICs), which were defined according to the World Bank Country Classification [13]. LMICs included low-, lower-middle-, and upper-middle-income countries. World geographical maps were built with the geographic information system QGIS (v3.10, https://qgis.org) to illustrate national COVID-19 deaths/1,000.000 children and CFR for children. Age-specific CFR and rate of ICU admission were calculated among the included cases with detailed age information. Microsoft Excel and Stata$^{®}$ v14.2 software were used for data synthesis. Outcomes were presented with 95% confidence intervals. Pearson's chi-squared tests were performed to compare outcomes between groups.

If a CFR or ICU admission rate could not be calculated because either the age range or dates of reporting of cases and deaths or ICU admission differed, or if only the absolute number of hospitalizations, instead of the number of cases, was reported, then we did not include these national data into our calculation of global CFR or ICU admission rate, but did include in the total number of deaths or ICU admissions. Data which may not be nationally representative, including case reports or case series from a specific hospital and subnational data (data from a specific city), as well as clinically diagnosed cases without PCR confirmation, were not

included in our primary calculations, but evaluated in the sensitivity analysis to investigate the robustness of our result. In order to ensure data accurately reflected the time frame between searches 2 and 3, we excluded countries with CDC COVID-19 levels 2 to 4 (moderate, high, very high transmission) that had not updated their national reports for more than 2 months prior to December 7 [14]. These national reports were evaluated in the sensitivity analysis.

## Results

A total of 28,557 articles from 16 databases were identified. After duplicates were removed, 16,027 records remained. Title and abstract screening excluded 13,562 records. The full text of 2,465 records were assessed in detail. 443 articles met the eligibility criteria (Fig 1), the characteristics of which are presented in S1 Table.

We also identified 225 national reports from governments, public health sectors, or national academic associations, of which 145 national reports met the inclusion criteria (Fig 1). The characteristics of the included national reports are presented in S2 Table.

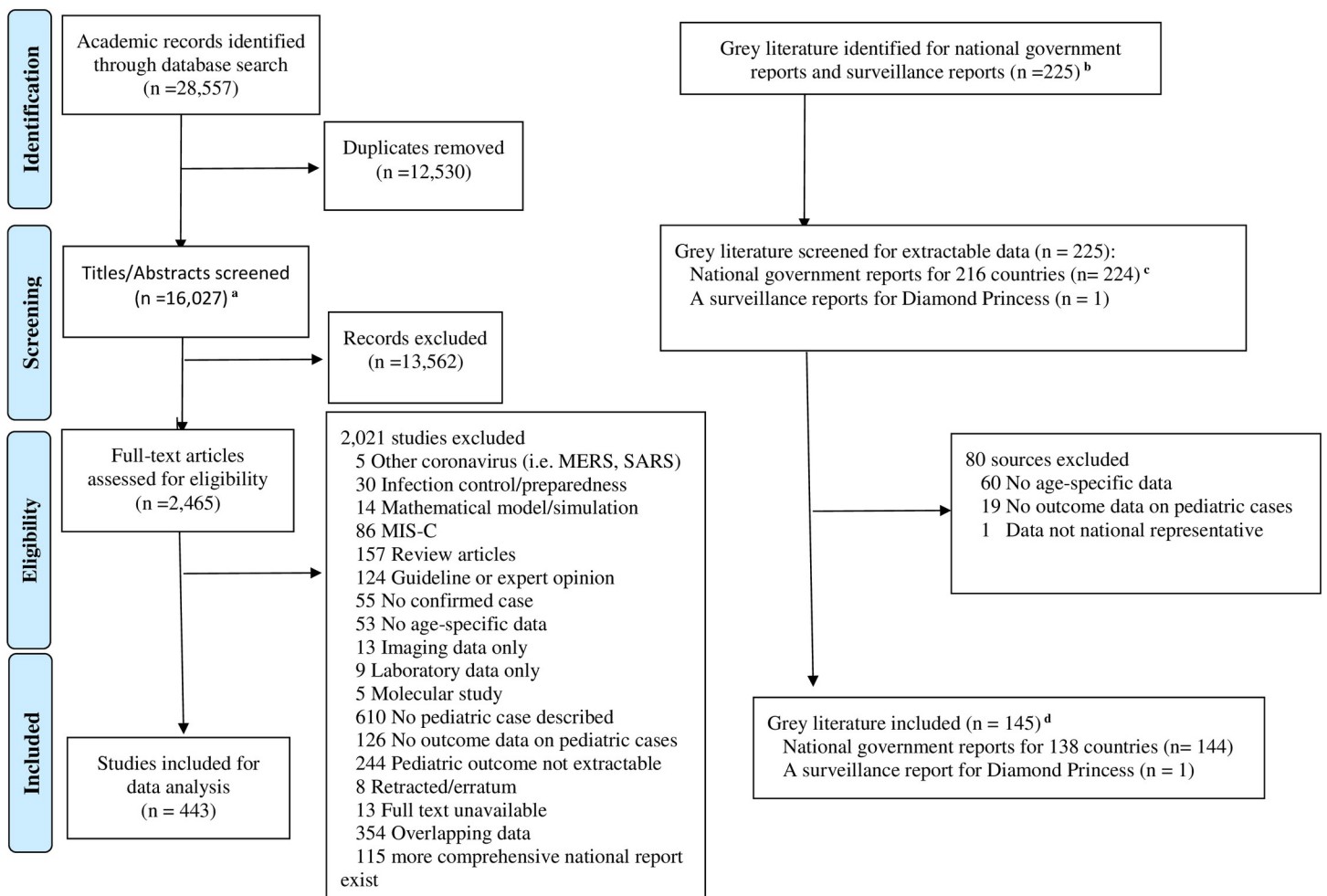

**Fig 1. PRISMA flowchart.** [a] Included two manually extracted article. [b] No national data from North Korea and Turkmenistan was identified. [c] Aggregated data from more than one eligible reports identified, screened, and included for data analysis/synthesis for 6 countries (Estonia, Namibia, Romania, Rwanda, United Kingdom, and the United States of America). [d] National reports from 8 countries (Burkina Faso, Cyprus, Guinea, Kazakhstan, Kenya, Oman, Pakistan, and Trinidad Tobago) were not included into primary analysis because they were reported more than 2 months before the date of final search (Dec 7, 2020), but they are included into the sensitivity analysis. MIS-C: Multisystem inflammatory syndrome in children.

The pediatric death and ICU admission data were available from national reports of 138 countries (59 from HIC and 79 from LMIC) and 22 countries (10 from HIC and 12 from LMIC), respectively. National reports from 8 countries had not been updated for more than 2 months before the date of final search (December 7, 2020). Eight countries did not report any confirmed COVID-19 cases in children as of December 7, 2020. Neither the number of pediatric deaths nor pediatric ICU admissions were available from 80 countries with confirmed SARS-CoV-2 infections as of the same date.

In total, there were 3,788 pediatric COVID-19 deaths and 3,118 ICU admissions identified from published literature or national reports worldwide. Among the 3,788 reported pediatric COVID-19 deaths, 321 (8.5%) and 3,394 (91.5%) deaths were reported from HIC and LMIC, while 16.5% and 83.5% of pediatric population from all included countries were from HIC and LMIC respectively. Of the 3,118 ICU admissions, 2,234 (71.7%) were from HIC, while only 720 (28.3%) ICU admissions were reported from LMIC. Fig 2 shows the variation of deaths/ 1,000,000 children by country. There is an excess of deaths in Middle and South American countries. Among countries where the nation-wide number of pediatric COVID-19 deaths or ICU admissions were available, COVID-19 deaths/1,000,000 children was calculated as 1.32 and 2.77 in HIC and LMIC ($p < 0.001$) respectively (Fig 3), while ICU admission/1,000,000 children were 18.80 and 1.48 in HIC and LMIC ($p < 0.001$), although the ICU admission data from lower MIC and LIC were scarce (S4 Table).

Overall, 3,379,049 children with a known outcome for fatality and 1,738,306 children with a known outcome for ICU admission were included in our calculation of CFR and ICU admission rate as a secondary outcome analysis. The estimated pediatric CFR was 0.061% (95% CI [0.059–0.064%]) (2,061/3,379,049) and pediatric ICU admission rate was 0.152% [0.146–0.158%] (2,644/1,738,306). The world map of national pediatric CFR is presented in S2 Fig. The pediatric CFRs in HICs, upper MICs, lower MICs and LIC were 0.012% [0.010–0.013%], 0.150% [0.140–0.162%], 0.433% [0.407–0.461%] and 0.241% [0.230–0.253%], respectively (S3

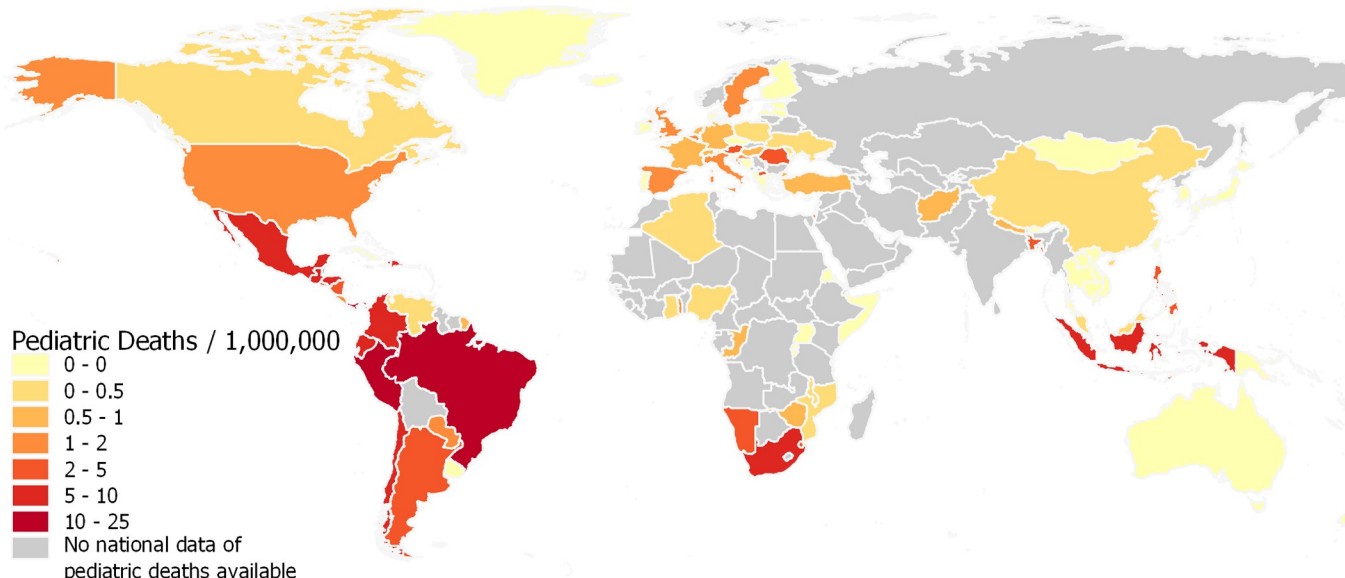

**Fig 2. World map of national pediatric COVID-19 deaths (/1,000,000 children).** The map was built with the geographic information system QGIS (v3.10, https://qgis.org) and the World Bank Official Boundaries Data Set (https://datacatalog.worldbank.org/dataset/world-bank-official-boundaries). Deaths are presented per million children. Countries of no pediatric case reported includes the country clearly report that there was no confirmed case in children in the national report as of December 7, 2020. National reports published more than 2 months before December 7 were included, if the countries were CDC COVID-19 Level 1 (low transmission) since the date of report.

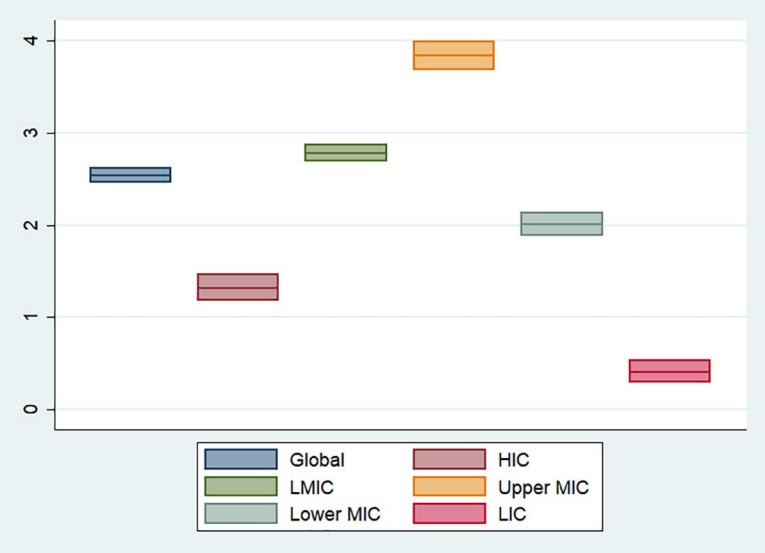

**Fig 3. Pediatric COVID-19 deaths (/1,000,000 children) by country income.** The ranges are presented by 95% confidence intervals of each proportion. Global includes all countries defined by World income. Abbreviations: HICs, high-income countries; LMICs, low- and middle-income countries; MICs, middle-income countries; LICs, low-income countries.

Fig). The pediatric CFRs are significantly higher in LMIC than HIC (CFR 0.29% [0.28–0.31%] in LMIC vs 0.03% [0.03–0.03%] in HIC; $p < 0.001$). ICU admission rates were 0.128% [0.122–0.133%] in HIC and 0.397% [0.367–0.430%] in upper MIC; $p < 0.001$).

Age-specific deaths and ICU admissions/1,000,000 children were 10.03 [9.07–11.07] and 16.84 [15.15–18.67] in < 1 year old, 1.64 [1.44–1.86] and 1.40 [1.16–1.67] in 1–4 years old, 0.92 [0.79–1.06] and 0.73 [0.58–0.90] in 5–9 years old, 1.13 [0.99–1.30] and 0.97 [0.79–1.12] in 10–14 years old and 2.70 [2.47–2.95] and 2.78 [2.47–3.12] in 15–19 years old, respectively (Fig 4 and S4 Table). Disaggregated age data were available in 558,961 and 242,827 children with confirmed SARS-CoV-2 infection as denominators for calculation of age-specific CFR and ICU admission rate, respectively. Age-specific CFR and ICU admission rate are shown in S3 Fig and S4 Table. Infants < 1 year old had the highest CFR (0.58% [0.50–0.67%]) and ICU admission rate (1.41% [1.27–1.56%]).

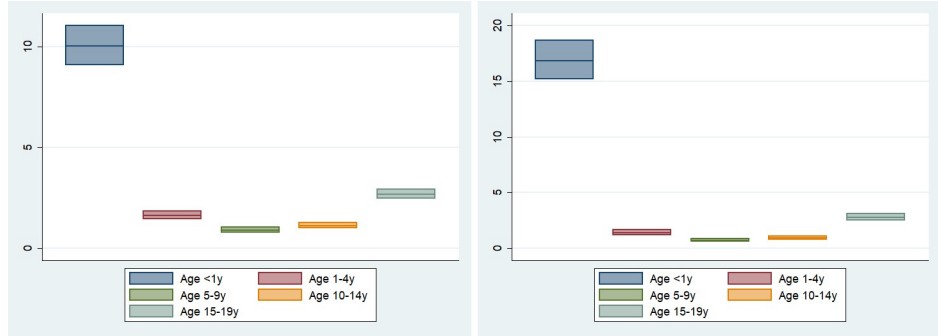

**Fig 4. Age-specific deaths and ICU admissions (/1,000,000 children).** A. Age-specific deaths (/1,000,000 children). B. Age-specific ICU admissions (1,000,000 children). Age-specific national data with up to one year difference of age buckets were included. For example, age-specific national data reporting outcomes of 1–5 years and 10–15 years were included in our calculation of 1–4 years and 10–14 years. Abbreviations: y, years.

Of 1,924 pediatric COVID-19 ICU admissions with data regarding required maximal respiratory support, 578 (30.0%) required invasive mechanical ventilation (S5 Table). Twenty-nine neonatal COVID-19 fatalities and 129 neonatal ICU admissions were identified. Neonatal case fatality rate and ICU admission rates were estimated as 1.74% (15/864) and 10.81% (91/842), respectively.

The result of one-way sensitivity analysis is presented in S6 Table). In all sensitivity analysis scenarios, differences in the deaths/1,000,0000 children and CFR between HIC and LMIC were maintained (deaths/1,000,0000 children in HICs = 1.26–1.32 and in LMICs = 2.43–2.77; $p < 0.001$ and CFR in HICs = 0.01–0.01% and CFR in LMICs = 0.23–0.24%; $p < 0.001$).

## Discussion

To our knowledge, this study is the largest and most comprehensive systematic review of severe pediatric COVID-19 outcomes to date. The fact that the majority of pediatric COVID-19 fatalities were reported from LMIC, and COVID-19 death using age-specific population as the denominator is greater in LMIC, indicates that the impact of pediatric COVID-19 fatality may in fact be larger in LMIC than that of HIC. The larger impact of pediatric COVID-19 fatalities in LMICs may be in part a consequence of a lower capacity or quality of healthcare system overall. The findings of larger impact of pediatric COVID-19 fatalities for LMICs is also consistent with the higher total all cause child death numbers in these countries [15, 16]. Although many child deaths in LMIC occur without medical attention [17], whether under-counting deaths in LMIC is also an issue for pediatric COVID-19 deaths is unknown. Our results have shown the opposite to be true for global pediatric ICU admissions. This is in part reflective of better case identification of severe cases of COVID-19 in HIC and dramatically better access to pediatric ICU level care, instead of the difference of disease severity in children [8, 18]. We also showed that the CFR was significantly higher in LMICs compared to HICs (0.24% and 0.01%; $p < 0.001$), a finding not previously reported, and one that may in part be related to less testing of children with less severe disease in LMICs. Of note, LICs report the lowest COVID-19 deaths/1,000,000. Given the fact that LICs still have higher CFR compared to HICs and upper MICs, this may reflect differing epidemiology of the outbreak situation of COVID-19 in LICs, rather than children in LICs being less likely to have severe COVID-19 once infected.

National CFRs can be influenced by many factors, including the age-distribution of the population, phase of the epidemic, and access to and capacity of the health care system [19–22]. Given that children are more likely to be asymptomatic or paucisymptomatic than adults [23], calculations of rates using number of infected children as a denominator are likely to be overestimates. For example, nationwide seroprevalence of SARS-CoV-2 in Spain ranged from 1.1–4.0% in children [24], which means the estimated number of children infected with SARS-CoV-2 from serological tests is much larger than the number of PCR-diagnosed cases in the nation. However, we believe that using serologically-diagnosed cases, instead of PCR diagnosed cases, as a denominator to calculate CFR or ICU admission rate is less appropriate because of variable access to this test across countries during the period under study as well as significant variability in type of serologic tests, and performance of these tests [25].

There are several limitations in our study. First, although our study also showed that among children, those younger than 1 year of age have the highest CFR and ICU admission rate (0.55% and 1.52%), many countries did not report disaggregated age-specific outcome data for children. Nevertheless, the rates we have found are higher than those previously reported [3–6]. Not all geographic regions equally reported national data in children. While many American and European countries reported nationwide data in children, the data from Africa and

Middle East were limited. The scarcity of national ICU admission data from lower MICs and LICs should also be noted. The large heterogeneity of included studies and national reports is another potential limitation of our study. Of note, because we specifically focused on fatality and ICU admission rate from acute COVID-19, the data regarding other important COVID-19 related outcomes, including MIS-C, as well as in-direct impacts of COVID-19 on child health such as disruption to routine childhood vaccination rates and the socio-developmental effects of school closures, were not evaluated in our study. While this study excluded known MIS-C cases, it is possible that some country's data may include these cases. Finally, our study could not assess the effect of underlying medical conditions, including malnutrition, on severe COVID-19 outcomes. Although our search found a few case reports from LMIC about fatal COVID-19 cases in children with malnutrition [26, 27], none of the study extracted in our systematic review investigated the impact of malnutrition on the severe COVID-19 outcomes in a population level.

## Conclusions

This systematic review is the first to document the global magnitude of impact, specifically fatality and ICU admission, for pediatric COVID-19. The results suggest that there may be a larger impact of pediatric COVID-19 fatality in LMICs compared to HICs. Given the lack of uniform testing strategies among children across the globe, temporal and geographic comparisons among children should be carefully undertaken in order to discern the possible extent of undiagnosed SARS-CoV-2-related deaths. While the true CFR from COVID-19 in children is likely to be lower than the numbers presented here due to limitations in counting total number of infections in children, these data can be taken as a starting point for comparing LMIC to HIC. Efforts should be made both within countries and by the global community to rigorously document age-specific outcomes in a timely manner as such data are important both to fully understand the global epidemiology and also to demonstrate where mitigation efforts, vaccine distribution and improving access to care are warranted.

## Supporting information

**S1 Checklist. PRISMA 2009 checklist.**
(DOC)

**S1 File. Supplementary methods.**
(DOCX)

**S1 Fig. World map of national pediatric case fatality rates.** Abbreviations: CFR, case fatality rate. CFRs are presented in percentages (%). Countries of no pediatric case reported includes the country clearly report that there was no confirmed case in children in the national report as of December 7, 2020. National reports published more than 2 months before December 7 are not included, if the countries are CDC COVID-19 Level 2–4 since the date of report.
(TIF)

**S2 Fig. Pediatric case fatality rate by country income.** The ranges are presented by 95% confidence intervals of each proportion. Global includes all countries defined by World income. Abbreviations: HICs, high-income countries; LMICs, low- and middle-income countries; MICs, middle-income countries; LICs, low-income countries.
(TIF)

**S3 Fig. Age-specific case fatality rate and ICU admission rate.** A. Age-specific case fatality rate. B. Age-specific ICU admission rate. Age-specific national data with up to one year

difference of age buckets were included. For example, age-specific national data reporting outcomes of 1–5 years and 10–15 years were included in our calculation of 1–4 years and 10–14 years. Abbreviations: y, years.
(ZIP)

**S1 Table. Study design and risk assessment for included articles from database search.**
(DOCX)

**S2 Table. National fatality and ICU admission data in children aged 0–19 years with confirmed SARS-CoV-2 infection (Dec 7–11, 2020).**
(DOCX)

**S3 Table. Study design and risk assessment for included articles from database search.**
(DOCX)

**S4 Table. Age-specific fatality and ICU admission (per 1,000,000 children), case fatality rate and ICU admission rate.**
(DOCX)

**S5 Table. ICU cases in children aged 0–19 years with confirmed SARS-CoV-2 infection cases who have known respiratory or circulatory support outcome.**
(DOCX)

**S6 Table. Sensitivity analysis using alternative data source.**
(DOCX)

## Acknowledgments

We thank Dr. Lorena Barra from The Hospital for Sick Children, University of Toronto, Ms. Hengameh Ghaedpur from The Hospital for Sick Children, University of Toronto, Mr. Zhang Jian, and Mr. Enis Hasim for reviewing Portuguese, Farsi, Chinese, and Turkish data.

## Author Contributions

**Conceptualization:** Taito Kitano, Mao Kitano, Carsten Krueger, Hassan Jamal, Hatem Al Rawahi, Rachelle Lee-Krueger, Rose Doulin Sun, Sandra Isabel, Marta Taida García-Ascaso, Hiromi Hibino, Bettina Camara, Marc Isabel, Leanna Cho, Helen E. Groves, Pierre-Philippe Piché-Renaud, Michael Kossov, Ikuho Kou, Ilsu Jon, Ana C. Blanchard, Nao Matsuda, Quenby Mahood, Anupma Wadhwa, Ari Bitnun, Shaun K. Morris.

**Data curation:** Taito Kitano, Mao Kitano, Carsten Krueger, Hassan Jamal, Hatem Al Rawahi, Rachelle Lee-Krueger, Rose Doulin Sun, Sandra Isabel, Marta Taida García-Ascaso, Hiromi Hibino, Bettina Camara, Marc Isabel, Leanna Cho, Helen E. Groves, Pierre-Philippe Piché-Renaud, Michael Kossov, Ikuho Kou, Ilsu Jon, Ana C. Blanchard, Nao Matsuda, Quenby Mahood, Ari Bitnun, Shaun K. Morris.

**Formal analysis:** Taito Kitano, Mao Kitano, Carsten Krueger, Hassan Jamal, Hatem Al Rawahi, Rachelle Lee-Krueger, Rose Doulin Sun, Sandra Isabel, Marta Taida García-Ascaso, Hiromi Hibino, Bettina Camara, Marc Isabel, Leanna Cho, Helen E. Groves, Pierre-Philippe Piché-Renaud, Michael Kossov, Ikuho Kou, Ilsu Jon, Ana C. Blanchard, Nao Matsuda, Ari Bitnun, Shaun K. Morris.

**Investigation:** Taito Kitano, Mao Kitano, Carsten Krueger, Hassan Jamal, Hatem Al Rawahi, Rachelle Lee-Krueger, Rose Doulin Sun, Sandra Isabel, Marta Taida García-Ascaso, Hiromi

Hibino, Bettina Camara, Marc Isabel, Leanna Cho, Helen E. Groves, Pierre-Philippe Piché-Renaud, Ari Bitnun, Shaun K. Morris.

**Methodology:** Taito Kitano, Mao Kitano, Carsten Krueger, Hassan Jamal, Hatem Al Rawahi, Rachelle Lee-Krueger, Rose Doulin Sun, Sandra Isabel, Marta Taida García-Ascaso, Hiromi Hibino, Bettina Camara, Marc Isabel, Leanna Cho, Helen E. Groves, Pierre-Philippe Piché-Renaud, Quenby Mahood, Anupma Wadhwa, Ari Bitnun, Shaun K. Morris.

**Project administration:** Taito Kitano, Mao Kitano, Carsten Krueger, Hassan Jamal, Hatem Al Rawahi, Rachelle Lee-Krueger, Hiromi Hibino, Bettina Camara, Marc Isabel, Leanna Cho, Helen E. Groves, Pierre-Philippe Piché-Renaud, Anupma Wadhwa, Ari Bitnun, Shaun K. Morris.

**Resources:** Taito Kitano, Carsten Krueger, Rose Doulin Sun, Sandra Isabel, Quenby Mahood.

**Software:** Taito Kitano.

**Supervision:** Anupma Wadhwa, Ari Bitnun, Shaun K. Morris.

**Validation:** Taito Kitano, Carsten Krueger, Hassan Jamal, Anupma Wadhwa, Ari Bitnun, Shaun K. Morris.

**Visualization:** Taito Kitano, Mao Kitano, Carsten Krueger, Rachelle Lee-Krueger, Sandra Isabel, Marc Isabel, Shaun K. Morris.

**Writing – original draft:** Taito Kitano, Mao Kitano, Carsten Krueger, Hatem Al Rawahi.

**Writing – review & editing:** Taito Kitano, Carsten Krueger, Hassan Jamal, Rachelle Lee-Krueger, Rose Doulin Sun, Sandra Isabel, Marta Taida García-Ascaso, Hiromi Hibino, Bettina Camara, Marc Isabel, Leanna Cho, Helen E. Groves, Pierre-Philippe Piché-Renaud, Michael Kossov, Ikuho Kou, Ilsu Jon, Ana C. Blanchard, Nao Matsuda, Quenby Mahood, Anupma Wadhwa, Ari Bitnun, Shaun K. Morris.

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
