## [Decision Letter · Decision Letter 0]

23 Nov 2020

PONE-D-20-31841

The differential impact of acute pediatric COVID-19 between high-income countries and low- and middle-income countries: A systematic review of fatality and ICU admission in children worldwide

PLOS ONE

Dear Dr. Kitano,

Thank you for submitting your manuscript to PLOS ONE. After careful consideration, we feel that it has merit but does not fully meet PLOS ONE’s publication criteria as it currently stands. Therefore, we invite you to submit a revised version of the manuscript that addresses the points raised during the review process.

This manuscript reflects an enormous amount of work and will provide an important contribution to current knowledge about pediatric morbidity and mortality worldwide. Please address all the comments by the two reviewers, who spent considerable time and effort reviewing the paper.

In addition, in the discussion section, please mention the geographic distribution of the manuscripts and data cited in the review, and highlight the fact, which is clear from the tables, that  much of the data from LMICs was from certain regions, such as the Americas, and other regions, like Sub-Saharan Africa and Asia, contributed very little data.

Finally, you mentioned your willingness to update your search to cover the period that has elapsed from the time of the last search until now. Because new data are constantly emerging in this pandemic, I believe updating the search to include the latest manuscripts and reports would make the findings of this review even more timely and relevant. I would encourage you to do this if you and your team are still willing and able.

We look forward to receiving your revised manuscript.

Kind regards,

Mark Katz

Academic Editor

PLOS ONE

Journal Requirements:

2. Please confirm that you have included all items recommended in the PRISMA checklist including:

-    a Supplemental file of the results of the individual components of the quality assessment, not just the overall score, for each study included.

-    See https://journals.plos.org/plosmedicine/article?id=10.1371/journal.pmed.1000100#pmed-1000100-t003 for guidance on reporting.

Thank you.

3. We note that Figure 2 and Figurere S1 in your submission contain map images which may be copyrighted. All PLOS content is published under the Creative Commons Attribution License (CC BY 4.0), which means that the manuscript, images, and Supporting Information files will be freely available online, and any third party is permitted to access, download, copy, distribute, and use these materials in any way, even commercially, with proper attribution. For these reasons, we cannot publish previously copyrighted maps or satellite images created using proprietary data, such as Google software (Google Maps, Street View, and Earth). For more information, see our copyright guidelines: http://journals.plos.org/plosone/s/licenses-and-copyright.

3.1.    You may seek permission from the original copyright holder of Figure 2 and Figurere S1 to publish the content specifically under the CC BY 4.0 license. 

3.2.    If you are unable to obtain permission from the original copyright holder to publish these figures under the CC BY 4.0 license or if the copyright holder’s requirements are incompatible with the CC BY 4.0 license, please either i) remove the figure or ii) supply a replacement figure that complies with the CC BY 4.0 license. Please check copyright information on all replacement figures and update the figure caption with source information. If applicable, please specify in the figure caption text when a figure is similar but not identical to the original image and is therefore for illustrative purposes only.

Reviewers' comments:

Reviewer's Responses to Questions

**Comments to the Author**

1. Is the manuscript technically sound, and do the data support the conclusions?

Reviewer #1: Partly

Reviewer #2: Yes

2. Has the statistical analysis been performed appropriately and rigorously? 

Reviewer #1: Yes

Reviewer #2: Yes

3. Have the authors made all data underlying the findings in their manuscript fully available?

Reviewer #1: Yes

Reviewer #2: Yes

4. Is the manuscript presented in an intelligible fashion and written in standard English?

Reviewer #1: No

Reviewer #2: Yes

5. Review Comments to the Author

Reviewer #1: This is an extremely ambitious and comprehensive analysis of the published data regarding mortality and ICU admission for PCR-confirmed SARS-Cov2 infection in children. The paper required a large amount of work and cooperation among the coauthors and as such represents an admirable effort. That said, the pandemic is still in full swing, many countries currently experiencing their second waves as the Northern hemisphere moves into winter; thus it must be considered an interim snapshot of the first 8 months of the pandemic only.

General comments:

There is a problem with both the terminology of the outcomes addressed and their validity. The outcomes addressed in this study include case fatality rates (CFR), ICU admission rates and "fatality prevalence". The latter is not an acceptable term in infectious disease epidemiology or general epidemiology for that matter. Death is an instantaneous event and therefore cannot be prevalent. It can only be incident. "Mortality rate" might be an acceptable alternative, "deaths from COVID per million" could also be used. The "period prevalence" of death is not pertinent, for similar reasons. In fact, in the absence of person-time denominators, the number of days since the declaration of the pandemic, or some other benchmark date ( eg Dec 31, 2019) should be taken into account when reporting these rates since they will be highly influenced by date of reporting. CFR, as the authors note, is highly dependent on testing policies and will inevitably rise where testing is scarce and limited to symptomatic or severely ill children, and therefore will be elevated in resource-scarce settings. The authors should consider "normalizing" these rates by tests per million in the populations studied.

Regarding ICU admission rates, this is clearly not a fair comparison between HIC and LMIC, many of which probably have no pediatric ICU care. Access to this type of care should be mentioned in the introduction. The dichotomization of the data into HIC and LMIC is questionable given the more detailed information presented in Figures 3A and B.

Specific Comments

Abstract – a definition of pediatric cases ( up to 19 years) should appear in the abstract). The percentage of deaths in HIC and LMICs should also be accompanied by percent of the world's children living in HICs and LMICs respectively. For instance, according to the World Bank 16% of the population in HIC and 27% in LMIC are children under the age of 14 –https://data.worldbank.org/indicator/SP.POP.0014.TO.ZS?name_desc=true. Do we know the total numbers up to age 19? The proportion dying of COVID-10 should also be presented in the context of the total proportion of children in these countries. The statement "The highest fatality prevalence and CFR were in infants < 1 year old (9.58/1,000,000 children and 0.55%, respectively) does not take into account HIC and LMIC – this comparison should be added.

Introduction- the authors acknowledge that previous studies have shown that death from COVID-19 in children is rare. They should stress what this study adds. Please mention ICU availability in LMICs to justify the pertinence of this comparison.

Methods:

Country reports are available to end of August, while published reports end as of August 10. Given delays in publishing, this means that published reports are highly skewed to the early months of the pandemic. In a revision, these should be updated. It is not clear how data overlap between country reports and published manuscripts was handled. Many case series are included – these may be outbreak investigations in schools, clinical case series – how were data extracted from these studies concerning the study outcomes? How do we know that there are no duplicate reports?

Results:

Lines 213-215 – in order to interpret these results, as mentioned, we need to know what proportion of children live in LMICS

Figure 3A shows variation by country- there is a clear excess of deaths in South America in countries around the equator. This is not addressed in the text. Supplement S2 B – there is no HIC in the graph although the abbreviation is found in the legend. Figure 3Bthe picture examining LIC LMIC HMIC and HIC is very different from the dichotomous analysis presented in the abstract and results.

Discussion line 282 – there are many reasons for a higher CFR besides less testing in LICs/ Is it possible that baseline characteristics of children admitted for COVID-19 differ between HIC and LIMIC? Could malnutrition, control of chronic conditions be playing a role in case fatality. This is important since comorbidity has been identified as a major risk factor for death in pediatric cases in HICs, according to multiple reports. Lines 283-300- this is a very long and repetitive explanation of the same point. Discussion can be shortened and tightened up.

Style points:

Line 94: why "thus"?

Line 147: missing period after "Bank"

Line 168: Sentence is unclear and needs rewriting. Note that sometimes data are plural in the manuscript and sometime singular. The former are preferable, but need to be consistent throughout.

Line 185: What are "proportional outcomes"?

Line 192: Representative not representatives

Line 231: "that of" should be deleted

Line 234: Age-specific number of fatality is not an accepted term

Line 244 – " data of" should be: data regarding or data pertaining to

Lines 250-253: Sentence needs to be rewritten for clarity

Line 297: tests not test. Following this there are several compound words missing hyphens serologically –diagnosed PCR-diagnoses, age-specific

Table 1 – Date should be added to the heading (to 31.08.2020)

Reference 13 in Supplement 2 – Israel is misspelled. (Isreal)

Reviewer #2: I would like to thank the authors for the immense work they did to conduct this systematic review. The topic of the review is obviously very important.

My comments:

Abstract:

Methods: To evaluate the magnitude of global ACUTE Covid-19 death, Acute does not seem correct, since not all PCR positive cases are necessary acute. They might also have been tested because were close contact of a case and were actually asymptomatic.

The number of reviewed articles and reports should be in the results not in the methods.

Since the Authors in the methods section report that only the first search round was performed on the countries national websites, I would suggest to remove the 11 languages from the abstract. Having the time period of the search in here would be very useful.

Conclusions: We knew that the COVID-19 an cause fatale disease in children, both in HIC and LMIC, and I would not say that should be the main conclusion of your study.

Introduction

Line 75 – the authors should update the references reported. New reports with child mortality are available

https://www.cdc.gov/mmwr/volumes/69/wr/pdfs/mm6937e4-H.pdf

https://services.aap.org/en/pages/2019-novel-coronavirus-covid-19-infections/children-and-covid-19-state-level-data-report/

https://www.thelancet.com/journals/lanchi/article/PIIS2352-4642(20)30177-2/fulltext

Row 79 – number of cases is also dependent on the size of the population

Row 80-83 – This seems more methods than introduction

Methods

Line 90-91 – Why primary case definition instead of just case definition?

Line 93-96 – What do you mean with POST-infectious multi-system inflammatory syndrome? What if the child had MIS-C during the hospitalization? I think the syndrome as such in COVID pediatric patients was not described from the beginning of the pandemic, therefore some of the children included in the review might have had it but was not clearly stated as such. How can the authors be sure that none of the included children had it?

Results

Line 207-209 – I am not sure I understand here the difference between countries or territories and then countries again. Could you please explain this sentence?

Line 215 – Instead of global I would suggest to use “reported” deaths

Line 225 – I would suggest to use estimated CFR and ICU admission rate instead of global

Line 250-252 – This seem to be more appropriate in the Methods part

Discussion

I find the discussion to be a little redundant, with a lot or results repetitions. Would prefer more reflections on the results and their interpretation. Like an interpretation of the results in figure 3A between UpperMIC and LIC.

Would also suggest to list the limitations of the study in a more ordered way instead of just spreading in the text.

Line 267-268 – I am not sure this is the case for Covid. Since there was high concern about the limited capacity of low middle income countries to perform PRC test, there has been high attention on the number of deaths, in all age groups so I would not conclude that there might have been many Covid related deaths which have been missed in the pandemic.

Line 269-271 – As above, there has been high attention to the number of deaths in low middle income countries, therefore I would not suggest that the prevalence of pediatric COVID-19 death is likely to be much higher

Line 293-295 – This is probably more true in LMIC than in HIC

Conclusion

Throughout the manuscript the authors talk about impact, burden and severe outcomes. Would be good to be consistent and align the terminology.

Line 326-335 – This sound more a continuation (and repetition) of the discussion more than a conclusion.

Figures

Table 1. I would suggest to put this in the supplement. Would also say to include the number of deaths and ICU admissions for each country

Figure 1. The resolution of the image is not good enough for me to read the content

6. PLOS authors have the option to publish the peer review history of their article (what does this mean?). If published, this will include your full peer review and any attached files.

Reviewer #1: No

Reviewer #2: No

---

## [Author Response · Author response to Decision Letter 0]

2 Jan 2021

Please see the file for "response to reviewer"

---

## [Decision Letter · Decision Letter 1]

15 Jan 2021

PONE-D-20-31841R1

The differential impact of pediatric COVID-19 between high-income countries and low- and middle-income countries: A systematic review of fatality and ICU admission in children worldwide

PLOS ONE

Dear Dr. Kitano,

Thank you for submitting your manuscript to PLOS ONE. After careful consideration, we feel that it has merit but does not fully meet PLOS ONE’s publication criteria as it currently stands. Therefore, we invite you to submit a revised version of the manuscript that addresses the points raised during the review process.

We look forward to receiving your revised manuscript.

Kind regards,

Mark Katz

Academic Editor

PLOS ONE

Additional Editor Comments (if provided):

Please add the dates of the analysis to the title of the manuscript.

Reviewers' comments:

Reviewer's Responses to Questions

**Comments to the Author**

1. If the authors have adequately addressed your comments raised in a previous round of review and you feel that this manuscript is now acceptable for publication, you may indicate that here to bypass the “Comments to the Author” section, enter your conflict of interest statement in the “Confidential to Editor” section, and submit your "Accept" recommendation.

Reviewer #1: (No Response)

Reviewer #2: All comments have been addressed

2. Is the manuscript technically sound, and do the data support the conclusions?

Reviewer #1: Yes

Reviewer #2: Yes

3. Has the statistical analysis been performed appropriately and rigorously? 

Reviewer #1: Yes

Reviewer #2: Yes

4. Have the authors made all data underlying the findings in their manuscript fully available?

Reviewer #1: Yes

Reviewer #2: Yes

5. Is the manuscript presented in an intelligible fashion and written in standard English?

Reviewer #1: Yes

Reviewer #2: Yes

6. Review Comments to the Author

Reviewer #1: The authors have taken careful note of the reviewers' comments and the manuscript is updated and improved. It could still benefit from some style and linguistic improvements eg "There is a geographical disparity about countries which contributed to our study." - this sentence is unclear - does this mean not all geographic regions were equally presented? Are the authors referring to the quality of data which varies according to geography. The paper should be carefully reread for clarity. The conclusions of the study perhaps should add a suggestion that given the lack of uniform testing strategies among children across the globe, careful temporal and geographic comparisons of all-cause mortality among children should be undertaken during the COVID-19 pandemic and periods before and after, in order to discern the possible extent of undiagnosed SARs-Cov2-related mortality.

Reviewer #2: I thank the authors for addressing all my comments and the extra work done in updating the research.

I find that the manuscript reads better now.

Minor suggestion:

- Discussion – (Line 378-9): “While this study focuses on acute COVID-19, it is possible that some country’s data may include cases of MIS-C.” Here I would suggest rephrasing in “While this study excluded MIS-C cases, it is possible that some country’s data may include these cases.”

7. PLOS authors have the option to publish the peer review history of their article (what does this mean?). If published, this will include your full peer review and any attached files.

Reviewer #1: No

Reviewer #2: No

---

## [Author Response · Author response to Decision Letter 1]

15 Jan 2021

Thank you very much for your time to review our revised manuscript. We sincerely appreciate editor and reviewers’ time to timely process and review our article.

Please see our response to reviewer file.

Thank you again for your time in this unprecedented time.

---

## [Editor Report · Decision Letter 2]

19 Jan 2021

The differential impact of pediatric COVID-19 between high-income countries and low- and middle-income countries: A systematic review of fatality and ICU admission in children worldwide

PONE-D-20-31841R2

Dear Dr. Kitano,

We’re pleased to inform you that your manuscript has been judged scientifically suitable for publication and will be formally accepted for publication once it meets all outstanding technical requirements.

Kind regards,

Mark Katz

Academic Editor

PLOS ONE
---

## [Editor Report · Acceptance letter]

21 Jan 2021

PONE-D-20-31841R2 

The differential impact of pediatric COVID-19 between high-income countries and low- and middle-income countries: A systematic review of fatality and ICU admission in children worldwide 

Dear Dr. Kitano:

I'm pleased to inform you that your manuscript has been deemed suitable for publication in PLOS ONE. Congratulations! Your manuscript is now with our production department. 

Kind regards, 

on behalf of

Dr. Mark Katz 

Academic Editor

PLOS ONE